# Response of Soybean Yield and Certain Growth Parameters to Simulated Reproductive Structure Removal

**Sarah Kezar [1], Anna Ballagh [2], Vanaja Kankarla [3], Sumit Sharma [2], Raedan Sharry [2] and Josh Lofton [2,*]**

1 Department of Soil and Crop Science, Texas A&M University, College Station, TX 77843, USA
2 Department of Plant and Soil Sciences, Oklahoma State University, Stillwater, OK 74078, USA
3 Department of Marine and Earth Science, Florida Gulf Coast University, Fort Myers, FL 33965, USA
* Correspondence: josh.lofton@okstate.edu

**Abstract:** Soybeans often encounter several in-season stressors that can alter retention of reproductive structures. To understand soybean response to structural losses through altered growth parameters—and, ultimately, yield—a field trial was established in Bixby, Oklahoma, in 2019 and 2020 and Perkins, Oklahoma, in 2019. Removal of reproductive structures occurred at full flower (R2), the beginning of pod development (R3), and the beginning of seed development (R5) and at three locations on the plant (top third (T), middle third (M), whole (W)). The impact of flower removal on yield at the R2 and R3 stages did not significantly differ from that in non-treated soybean. Pod removal as late as R5 from the upper fruiting positions (T) had a lesser impact on overall yield, with R5:T showing a reduction in seed number of 860 seeds plant$^{-1}$, whereas R5:M was 1921 seeds plant$^{-1}$ below the non-treated soybean. The middle portion of the mainstem was the location where the loss demonstrated was paramount at R5, as this region is a large sink and major contributor to yield. Late-season, stress-negating yield recovery, depending on the severity, may indicate that management practices should anticipate physiological limitations for stress, as well as the potential for relative yield recovery and yield improvement.

**Keywords:** soybean; yield recovery; stress; pod removal





## 1. Introduction

Currently, global soybean production takes place across approximately 75.5 million hectares, equating to nearly 6% of the world's arable farmland, with the United States being the leading global soybean producer [1]. Soybean grain yield is defined as the average mass of individual seeds produced by the mean number of plants per unit land area, with seed number per unit land area being a fundamental yield factor [2]. Maximum soybean yield potential has been reported to range from 7250 to 11,000 kg ha$^{-1}$ but such yields only exist in the absence of stress from planting to maturity [3,4]. As soybean is one of the five most relied upon global crops, it is important to understand the risks associated with soybean response to stressors throughout the growing season to protect profitability for producers and global food and feed security [5].

Soybean can often encounter several abiotic and biotic stressors throughout the growing season that alter the number of retained reproductive structures and, ultimately, lower yield. The challenges of abiotic and biotic stressors can include adverse weather, insect and weedy pests, disease, and diminished soil quality, which result in yield gaps in soybean production. As the loss of reproductive structures is a great challenge to overcome when soybean plants undergo stress periods, it is critical to understand how the soybean plant responds to loss of reproductive structures, as well as how and to what degree yield can be recovered. In essence, yield depends on the number of flowers a soybean plant produces during an average flowering period of 20 to 40 days—although it may be long as 90 days—but most flowers are produced within a narrower window [6–9]. The number

of flowers produced during the concurrent vegetative growth and first and full flowering stages (R1 and R2, respectively) equates to the yield and, inversely, the number of aborted flowers during this time directly limits the yield [10,11]. The R1 growth stage is noteworthy as this is the beginning of the transition from vegetative growth to reproductive growth. During this time, the flowers that develop on the nodes established previously mature into the reproductive organs that determine final seed number and, ultimately, yield [2,12].

Early-developing reproductive structures—pistils and stamens—are the primary plant structures at risk of being impaired by stress during anthesis [13], while ovule function is more sensitive to stress than pollen production [14]. Soybean flowers are at the greatest risk of abscission when stress occurs during soybean flowering as a consequence of resource competition [15]. However, Shaw and Laing [16] observed that stress at R1 allowed later recovery of pods when stress was alleviated before the end of R2. This potential for yield recovery can be attributed to the highly asynchronous nature of soybean flowering [17]. The flowering periods of indeterminate types are more extended compared to determinate types, but overall flower production is concentrated in a similar window amongst indeterminate and determinate soybean [18,19]. The ability of indeterminate soybean to produce flowers for an extended period of 20 to 30 days entails the potential to recover yield under stressful conditions, such as moisture and temperature stress [20,21]. The duration and timing of the flowering period are important factors relating to yield, as recovery from a stress event or abortion of flowers or pods can be mitigated by production of enough extra flowers [2,17].

The locations of node-bearing reproductive structures in a soybean plant influence the order of flower development, which oftentimes begins within nodes on the main raceme and then extends to secondary and tertiary branches and finally to sub-branches [2,22]. Pods from early flowers initiated on lower nodes are sinks that consume most of the assimilate supply and leave limited resources for late flowers developing simultaneously at the top of the mainstem and distal branch positions [17,23]. Consequently, later-developing pods towards the top of the soybean plant and outer branches also have a decreased chance of survival because of inadequate assimilate supply [23]. Assimilatory capacity—or source strength—affects branch and node numbers, as well as the numbers of pods per reproductive node on mainstems and branches, from the beginning of flowering to the beginning of pod development (R3) growth stages [12]. The impact of moisture stress at R3 limits biomass, restricting the creation of additional nodes that would supply yield and, at the same time, influencing the survival of late-developing flowers and pods through competition with larger sinks for resources [24].

During soybean seed fill—or the R5 growth stage—no additional flowers are produced to mitigate losses from aborted pods and low seed weight [25]. Environmental potential for pod and seed set does not require conditions for high levels of photosynthesis during the entire reproductive period, but reductions in photosynthesis and assimilate supply cannot be tolerated by the soybean plant for more than 14 days without a negative impact on seed number [26]. Seed weight has been found to have an inverse relationship with increasing levels of soil moisture stress and the potential to reduce seed numbers by 45% and limit the duration of the seed filling period, subsequently resulting in small and shriveled seeds [27]. Previous research has also identified restriction of water from the end of pod development to pod fill—or early R5—as a major limiting factor influencing soybean yield reduction [25,28]. Thus, stress during R5 brings major risks for yield loss as aborted pods directly influence the major yield components seed number and seed weight [28,29].

Remobilization of photosynthetic assimilates of C and N from leaves and vegetative material—the source—makes important contributions to the developing seeds—the sink—during R5 [30]. When demand for these nutrients exceeds supply, photosynthesis becomes limited, and senescence begins prematurely [31]. While excessive heat events aborting pods during R3 can be compensated for by increasing the seed weight of the remaining pods, heat stress events during R5 disrupt seed development with no remaining flowering window for yield recovery [21,32]. Crops experiencing late-season stresses are further prone to reductions in yield quality, with shriveled seeds or seeds exhibiting lower quality

properties due to an inadequate seed filling period [17,33]. Unfavorable environmental conditions can quickly compound unfavorable and uncontrollable abiotic stress events in soybean production [34].

While the information above highlights the potential loss of reproductive structures due to in-season stressors, limited research is currently available on the response of the plant and potential yield recovery when this occurs. Therefore, the objective of this study was to estimate the potential for soybean yield recovery from flower and pod removal from the top third of the mainstem (T), the middle third (M), and across the whole mainstem (W) during the R2 (full bloom), R3 (beginning of pod development), and R5 (beginning of seed development) growth stages.

## 2. Materials and Methods

The information provided in this article is partially associated with a chapter in a master's thesis produced by the authors at Oklahoma State University [35]. An additional year's worth of field data were collected and published in this article that were not in the thesis.

Field trials were conducted at the Mingo Valley Research Station (35°57′52.3″ N, 95°51′38.6″ W) near Bixby, OK, in 2019 and 2020 and the Cimarron Valley Research Station (35°59′09.1″ N, 97°02′47.1″ W) near Perkins, OK, in 2019. The soils at the Bixby and Perkins locations were both Mollisols. The soil series in Bixby was a mix of Wynona silty clay (fine-silty, mixed, active, thermic Cumulic Epiaquolls) and a Mason silt loam (fine-silty, mixed, active, thermic Pachic Argiudolls). The Perkins soil consisted of a Teller fine sandy loam (fine-loamy, mixed, active, thermic Udic Argiustolls). Rainfall in the 2019 growing season between May and October amounted to 808 mm and 865 mm in Bixby and Perkins, respectively (Figure 1a). In 2020, Bixby received 925 mm of rainfall and was the sole established location due to planting failure at Perkins resulting from early-season flood conditions (Figure 1a). Temperature patterns were similar across all site years (Figure 1b).

The soybean cultivar developed by Pioneer (Pioneer P48A6OX; Pioneer; Johnston, IA, USA), an indeterminate variety of maturity group IV, that contained the RoundUp Ready Xtend trait was planted in each site year. Plots were established using a custom built Monosem (Monosem Inc.; Edwardsville, Kansas) planter across site years. Before planting, the seeds were treated with Vault SP for Soybeans Inoculant (Vault SP; BASF; Ludwigshafen, Germany) to supplement the development of the bacterial component in the soybean root nodules, *Bradyrhizobium japonicum*. The planting populations were 260,000 seeds ha$^{-1}$ at Perkins in 2019 and 308,881 seeds ha$^{-1}$ at Bixby in 2019 and 2020, with planting densities and dates reflecting nearby producer practices (Table 1). All trials used 76 cm row spacing. Seeding depth was targeted at 2.5 cm; however, this varied based on seedbed conditions.

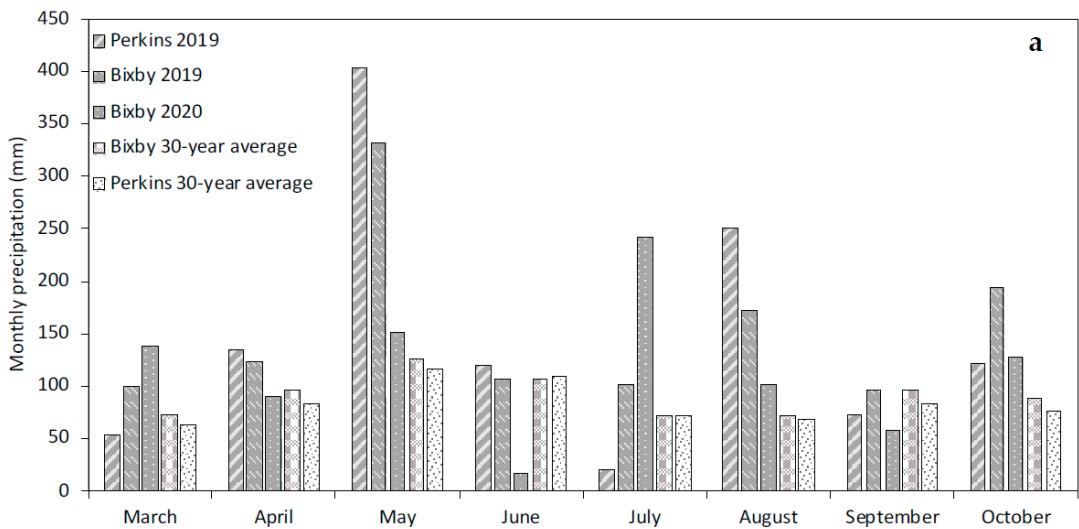

**Figure 1.** *Cont.*

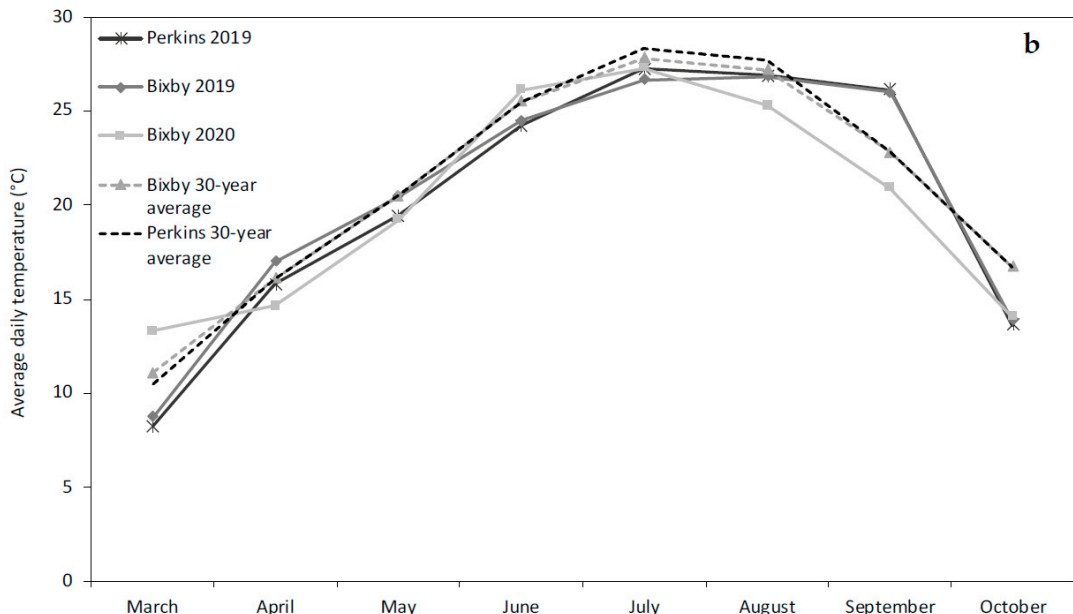

**Figure 1.** (**a**) Monthly precipitation (mm) received during the growing season at the Bixby field site in 2019 and 2020 and the Perkins field site in 2019. (**b**) Average daily temperature (°C) during the growing season for the Bixby field site in 2019 and 2020 and the Perkins field site in 2019.

**Table 1.** Planting population, planting date, and harvest date for field trials conducted in Perkins, OK, in 2019 and Bixby, OK, in 2019–2020.

| Location | Planting Population | Planting Date | Harvest Date |
|---|---|---|---|
| Perkins 2019 | 260,000 seeds ha$^{-1}$ | 15 May | 12 September |
| Bixby 2019 | 308,881 seeds ha$^{-1}$ | 21 May | 28 October |
| Bixby 2020 | 308,881 seeds ha$^{-1}$ | 21 May | 16 October |

Pre-plant application of pyroxasulfone (240 mL ha$^{-1}$) and glyphosate (1541 g ha$^{-1}$) was implemented immediately prior to planting for all locations and years. For in-season soybean management, post-emergent herbicide application of 3082 g ha$^{-1}$ of the active ingredient glyphosate and 1541 g ha$^{-1}$ of the active ingredient dicamba was implemented across site years, with herbicide rates reflecting optimal weed control. For control of in-season pests across site years, primarily southern green stinkbugs (*Nezara viridula*) and soybean pod worm (*Helicoverpa zea*), 1169 g ha$^{-1}$ of chlorantraniliprole (Prevathon; FMC Corporation; Philadelphia, PA) and 584 g ha$^{-1}$ of lambda-cyhalothrin, chlorantraniliprole (Besiege; Syngenta; Basel, Switzerland) were applied. Desiccation across site years was undertaken using gramoxone, which was applied at 1169 g ha$^{-1}$ two weeks before the respective harvest dates (Table 1). No phosphorous or potassium fertilizer were added to Bixby soils in 2019 and 2020 based on soil test recommendations. Fertilizer was applied to Perkins soils in 2019 with 44 kg ha$^{-1}$ P and 44 kg ha$^{-1}$ K via triple super phosphate (0-46-0) and muriate of potash (0-0-60).

The field study was arranged in a randomized complete block design with three soybean growth stages denoted R2, R3, and R5 and three locations for reproductive structure removal: the whole mainstem (W), the middle third (M), and the top third (T). Plots measured 3.1 m by 0.8 m (3 m$^2$). A smaller plot area was selected in order to obtain a more homogeneous region to evaluate the treatments. In addition to the treatment plots, a similarly sized plot was utilized as a non-treated control, and the reproductive structures were not altered. Across all site years, all treatments were replicated four times, leading to 40 (ten treatments replicated four times) plots in total at each location.

To target reproductive structure removal areas, soybean mainstem nodes were counted on representative plants before the R2, R3, and R5 treatment stages, respectively. For cumulative treatment location accuracy, the average number of nodes per soybean plant was used to divide the mainstem according to approximate mainstem node locations, and marking tape was tied to divide the soybean mainstem into M and T sections (Figure 2a,b). For the W treatment, reproductive structures were removed across the entirety of the mainstem and marking tape was thus not required.

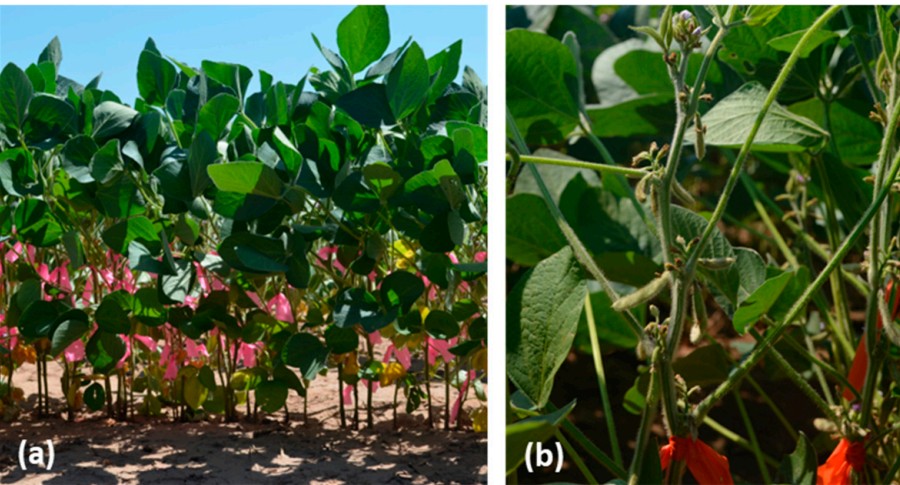

**Figure 2.** (**a**) Division of the middle portion of the mainstem with marking tape to identify locations for the removal of reproductive structures and (**b**) division of the top portion of the mainstem, depicted here at the R3 stage.

At the time of harvest, a subsample of three plants was collected from each treatment level within each plot, as well as from the NTC plots. These plants were utilized to quantify the average plant height (height), number of mainstem nodes (node), seed count plant$^{-1}$, and number of seeds within each pod. The number of pods located on the plant mainstem versus on the branches (*MVB*) was also of interest for the evaluation of the physiological response to the treatments. The MVB measurement was calculated by subtracting the number of pods located on the branches from the number of pods located on the mainstem, then dividing the outcome by the number of pods located on the branches as described here:

$$Mainstem\ versus\ branch\ pods\ (MVB) = \frac{(Branch - Mainstem)}{Branch}$$

A positive number indicated that more pods were located on the branches, while a negative number of pods signified that pod numbers were concentrated on the mainstem, and the findings were analyzed accordingly. It is important to note that this parameter does not give any indication of the number of pods produced, and a near-zero estimate for the MVB does not mean that lower numbers of pods have developed but that there are near equal numbers being produced on the mainstem and branches.

All soybean plants within each treatment row and NTC row were hand-harvested and threshed using a Kincaid thresher (18″ Heavy Duty Bundle Thresher; Kincaid; Haven, KS, USA). Plot weights were used to estimate yield on a per hectare basis. Thereafter, an electric seed counter (Electric Counter; Model 8502; The Old Mill Company; Savage, MD, USA) was used to count the number of seeds (SN) from each treatment area. In 2019 specifically, it was noted that selected treatments developed green stem syndrome before harvest. As a measure of percent greenness at harvest—or delayed senescence—Canapeo (Canapeo App; Oklahoma State University; Stillwater, OK, USA) was used to determine the percent green canopy remaining for each treatment at both locations prior to harvest in 2019.

Data were subject to ANOVA utilizing the Procedure Mixed function in SAS v.9.4 (SAS Institute Inc., Cary, NC, USA). The stages of simulated soybean stress (main plot) and cumulative treatment location (sub-plot), as well as their interactive effects, were designated as fixed variables in the model, whereas replication and site were considered random effects. Preliminary analysis determined that there were no significant differences between the Bixby and Perkins sites across years; therefore, sites were combined within each year of the study. For significant treatment effects as shown by ANOVA, means were separated at $p \leq 0.05$ with Fisher's protected LSD. Prior to statistical analysis, tests of normality and homogeneity of variation for data normality were performed in JMP (JMP®, Version X. SAS Institute Inc., Cary, NC, USA, 1989–2022). The assumptions of both normality and homogeneity of variance were met and, thus, no data transformation was required.

## 3. Results and Discussion

### 3.1. Crop Yield and Harvest Efficiency

Soybean yield potential from 2019 to 2020 did not vary between the non-treated control plants (NTC), which produced 4875 kg ha$^{-1}$ and 4728 kg ha$^{-1}$, as well as 7848 seeds plant$^{-1}$ and 5637 seeds plant$^{-1}$, in 2019 and 2020, respectively (Figure 3b,d). Across treatments in 2019 and 2020, there was a slight reduction in seed numbers in 2020 compared to 2019 but a lesser difference in yield, which could have been bridged by higher seed weight in 2020 (Figure 3b,d). This response could have been due to variable rainfall patterns and higher temperatures in the month of June in 2020 impacting the overall number of seeds produced, while no severe environmental stress was observed during the seed filling period later on in the season (Figure 1a,b). Limited precipitation in soybean production systems is certainly a challenge that, when coupled with historically high temperatures, can impact crop performance at all reproductive stages [14,16,36,37]. In this study, the effects of the interactions between stages (S; R2, R3, R5) and locations (L; middle, top, whole) on the crop performance parameters yield and seed number were of interest (Table 2).

**Table 2.** ANOVA table for main effects of and interactions between stage, location, and position for each observation in 2019 and 2020. Critical *p*-values indicate the main effects and interactions.

| *Source of Variation (2019)* | Yield | Plant Height | Nodes Plant$^{-1}$ | Seed Number Plant$^{-1}$ | Pods Plant$^{-1}$ | One Seed Pod | Two Seed Pods | Three Bean Pods | Four Bean Pods | Change in Pod Location | % Greenness at Harvest [a] | Pod Location [b] |
|---|---|---|---|---|---|---|---|---|---|---|---|---|
| S | <0.0001 | 0.054 | 0.3908 | <0.0001 | 0.1251 | 0.1353 | 0.2485 | 0.0689 | **0.0015** | **0.0017** | <0.0001 | <0.0001 |
| L | <0.0001 | 0.05 | 0.551 | <0.0001 | 0.0791 | 0.5861 | 0.2116 | **0.0296** | **0.0005** | 0.0004 | 0.0005 | 0.0038 |
| S*L | <0.0001 | **0.0064** | 0.7757 | **0.0002** | 0.4886 | 0.7528 | 0.3778 | 0.6181 | **0.0106** | **0.0089** | <0.0001 | 0.3645 |
| P | | | | | | | | | | | | <0.0001 |
| S*P | | | | | | | | | | | | <0.0001 |
| L*P | | | | | | | | | | | | 0.0049 |
| S*L*P | | | | | | | | | | | | <0.0001 |

| *Source of Variation (2020)* | Yield | Plant Height | Nodes Plant$^{-1}$ | Seed Number Plant$^{-1}$ | Pods Plant$^{-1}$ | One Bean Pod | Two Bean Pods | Three Bean Pods | Four Bean Pods | Change in Pod Location | % Greenness at Harvest [a] | Pod Location [b] |
|---|---|---|---|---|---|---|---|---|---|---|---|---|
| S | <0.0001 | 0.4546 | 0.7141 | <0.0001 | <0.0001 | <0.0001 | <0.0001 | <0.0001 | 0.8969 | <0.0001 | NA | <0.0001 |
| L | <0.0001 | 0.0562 | 0.005 | <0.0001 | 0.0074 | <0.0001 | 0.3625 | <0.0001 | 0.2178 | 0.0029 | NA | <0.0001 |
| S*L | <0.0001 | 0.5959 | **0.0138** | <0.0001 | 0.0009 | 0.9084 | <0.0001 | 0.3397 | 0.2676 | <0.0001 | NA | <0.0001 |
| P | | | | | | | | | | | | <0.0001 |
| S*P | | | | | | | | | | | | <0.0001 |
| L*P | | | | | | | | | | | | 0.0043 |
| S*L*P | | | | | | | | | | | | <0.0001 |

Note for source of variation. S = reproductive stage (R2, R3, R5); L = location of flower and/or pod removal (middle third, top third, whole mainstem); P = position on soybean plant (branch, mainstem). [a] % Greenness at harvest was only recorded in 2019. [b] Pod location was the only observation to include P (position) as a main effect or interactive effect.

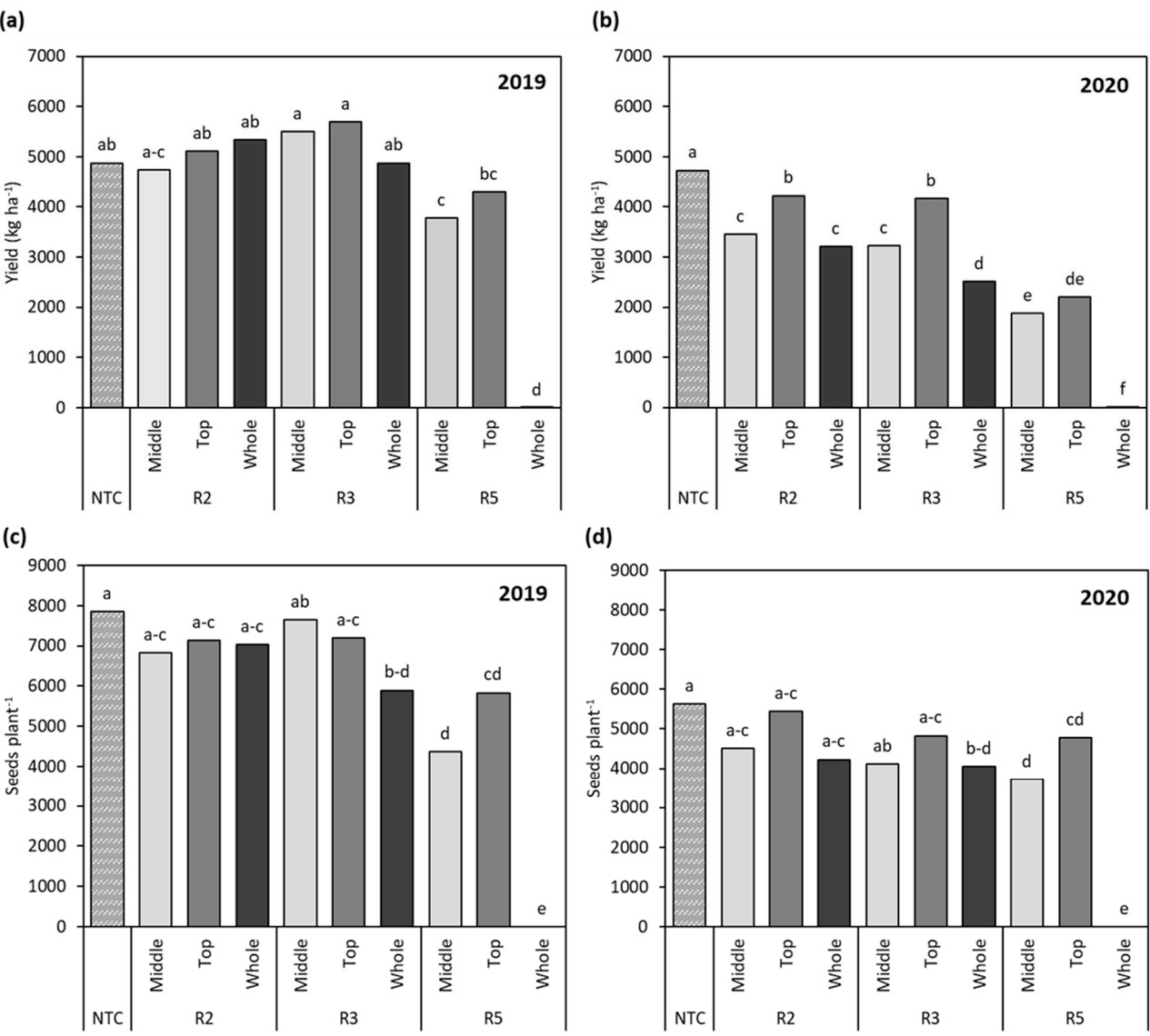

**Figure 3.** (**a**) Comparison of soybean yield as affected by cumulative stage and location effects in 2019 and (**b**) in 2020. (**c**) Comparison of numbers of seeds per plant as affected by cumulative stage and location effects in 2019 and (**d**) in 2020. Different letters indicate significant difference. NTC—non-treated control, R2—full flower, R3—beginning of pod development, R5—beginning of seed development.

Soybean flowering period was estimated to occur over a period of 20 to 40 days and yield was a function of the total number of flowers that developed and were maintained [7]. It is notable—but not surprising—that later flowering nodes or nodes on branches allowed reproductive recovery of R2:M at 4741 kg ha$^{-1}$, R2:T at 5112 kg ha$^{-1}$, and R2:W up to 5340 kg ha$^{-1}$, showing no significant difference from the NTC yield of 4875 kg ha$^{-1}$ in 2019 (Figure 3a,b). The ability of soybean to recover from simulated stress during early flowering demonstrated that yield can be compensated for at R2 through the occurrence of soybean production stressors. The soybean growth stage R2 is a unique time supporting both biomass and flower production in indeterminate soybean and full flower production in determinate soybean, marking the need for peak moisture uptake [13,16,38]. While the soybean variety utilized in this study was an indeterminate type, it should be noted that both indeterminate and determinate soybean types have a similar flowering window in which approximately 80% of flowers are produced [19]. The main difference between

indeterminate and determinate soybean varieties is that determinate types have a shorter overall timeframe for flowering outside of the concentrated flowering window [19].

The probability of abscission increasing with more distal nodal positions is well-documented [39]. More specifically, insufficient water supply due to environmental stress events at R2 results in a barrier in photosynthetic assimilate supply to support early embryonic development [14,15]. Our findings for yield recovery at R2 in 2020 were in agreement with this, as yields dropped by 1278 kg ha$^{-1}$ and 1519 kg ha$^{-1}$ when pod stress was simulated at R2:T and R2:W mainstem positions compared to the NTC (Figure 3b), and this period of the growing season received the lowest rainfall in the study (Figure 1b). As R2:W represented full removal of reproductive structures early in the season, these data illustrate how the plants were able to recover yield (Figure 3a,b) and seed numbers (Figure 3c,d) through a later flowering window in response to this early-season stress. When flowers were removed at R2:T in 2020, yield recovery was more attainable under limited moisture conditions, as only 507 kg ha$^{-1}$ of yield and 708 seeds plant$^{-1}$ were lost compared to the NTC (Figure 3b), aligning with previous findings that the upper third of the soybean plant is one of the most productive regions and the last of the floral organs to develop [40,41]. An extended flowering period has been shown to provide opportunities for yields to reach 6000 kg ha$^{-1}$, whereas a delayed flowering period limits yields from surpassing the 4500–5500 kg ha$^{-1}$ mark, illustrating the ability of the flowering period to facilitate yield recovery in close association with management practices [20].

The primary objective of the plant during the R3 growth stage is to supply assimilates to pods from early flowers [23]. The locations of flowers on the soybean plant influence flower development order from bottom to top along the mainstem; as such, the development of pods begins on the lower nodes, which are large sinks at this reproductive stage [17,41]. The plant simultaneously fills seeds on lower nodal positions while upper nodes are still within the flowering period and transitioning to pod development [7,32]. The dynamics whereby reproductive structure development is disrupted when stress is experienced at different locations on the plant ultimately play a large role in yield (Table 2 and Figure 3a,b). Although the top mainstem region is prone to abscission when stress is experienced, the loss of these developing pods at R3:T led to the highest yield of 5695 kg ha$^{-1}$ in 2019 (Figure 3a). This yield recovery can be compared to the findings of Spollen et al. [39] that the probability of abscission of upper nodes decreased when middle-node reproductive structures were removed. The mechanism underlying intra-raceme competitive ability remains unknown but could be related to differences in the time of floral initiation and remobilization of photosynthate from this reduction of sink strength [24,42]. Considering that R3:T in 2019 was the highest yielding treatment amongst R3 treatments but had the median seed number, we hypothesized that yield recovery was a result of remobilization of photosynthates increasing the yield component seed weight (Figure 3c). Removing pods at R3:M resulted in a comparable yield of 5493 kg ha$^{-1}$ in 2019 and the highest seed number of 7642 seeds plant$^{-1}$ (Figure 3a,c), and this yield recovery potential could have arisen from the pods on the middle portion of the mainstem having a proximal advantage in receiving assimilate supply [22]. However, this location advantage did not hold true in 2020 for R3:M when there was limited assimilate supply to support yield recovery, resulting in a yield reduction of 3226 kg ha$^{-1}$ and 4108 seeds plant$^{-1}$ as compared to R3:T, which demonstrated 4170 kg ha$^{-1}$ and 4830 seeds plant$^{-1}$ (Figure 3b,d). This agrees with the findings of Spollen et al. [39] that the lower portion of the soybean canopy is also the party most prone to reproductive losses. Moreover, the significant differences in yields and seed numbers between the R3:M, R3:T, and R3:W treatments indicate a differential response to the impact of the area where soybean stress is experienced and the extent of that stress (Figure 3b,d).

At R5, no additional flowers were produced to mitigate yield losses [25]. Simulated stress at R5:T in 2019 and 2020 had the least impact on yield, with a 582 kg ha$^{-1}$ decrease in 2019 and a yield of 2525 kg ha$^{-1}$ compared to the respective NTC (Figure 3a,b). Seed weight growth was assisted through mobilization of photosynthates from the upper leaves to

support the remaining developing pods at the middle and top of the mainstem [42]. In terms of the total number of seeds available for filling on the soybean plants, stress on the upper fruiting positions had a lesser impact on overall yield contributions in terms of the yield qualities that bring dividends at the elevator. This was demonstrated by the removal of pods at R5:T, which reduced seed numbers by 2018 seeds plant$^{-1}$ and 860 seeds plant$^{-1}$ in 2019 and 2020, respectively, and R5:M, which decreased seed numbers by 3484 seeds plant$^{-1}$ in 2019 and 1921 seeds plant$^{-1}$ in 2020 compared to the NTC (Figure 3b,d). Removal of pods at R5:M additionally led to decreases of 516 kg ha$^{-1}$ in 2019 and 329 kg ha$^{-1}$ in 2020 compared to R5:T soybean (Figure 3a,b). Pods located at the middle of the mainstem are known to be high contributors to yield due to their high pod-setting ratio and the increased numbers of seeds per pod, explaining the yield gap for pods removed from the middle of the mainstem compared to pods on the top portion [22,27]. The yield difference between the middle and top portions of the mainstem in terms of seed weight could be attributed to decreases in seed-filling duration when moving from lower nodes to each upper node position [32].

There was a different soybean response when pods were removed from across the whole of the mainstem for R5:W. Extensive stress at R5:W significantly impacted yield in terms of seed number, with no chance of reproductive recovery [25] (Figure 3a,b). As this seed-filling stage requires high water uptake to support seed fill and should increase proportionally with producer yield goals, assimilates that sequester seed growth can be easily limited by moisture and heat stress [4]. Furthermore, in 2019, the late-season reduction in pod load resulted in green stem, a physiological response to this alteration of the source–sink ratio. The removal or loss of pods—and, thus, a reduced sink—slow the movement of C and N from the vegetative tissue to pods as the stress response favors the source [43,44]. The occurrence of green stem prevents vegetative tissue from exuding moisture through the accumulation of photosynthetic assimilates of starch and N [43]. The impact of stress on the percentage of retained vegetative tissue at the time of harvest increased percent greenness by 4.5% for R5:M, 5.2% for R5:T, and 19.5% for R5:W compared to the standard physiological maturity shown in the NTC (Figure 4). This vegetative tissue did not reach physiological maturity at either site in 2019, despite desiccant application and frost (Figure 4), and plants were unharvestable due to this reaction to intolerable stressors. Late-season stressors that soybean producers can be subjected to, such as insects, disease, moisture stress, the environment, and hail, and that influence the source–sink ratio have this capability to produce green stem [45–48]. The potential for the onset of green stem due to these stressors certainly signifies the importance of late-season soybean management for optimal production [49].

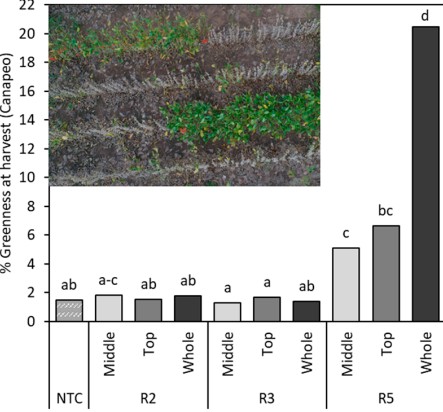

**Figure 4.** Canapeo readings prior to harvest indicating percentage of remaining vegetative tissue with green stem syndrome. Drone image depicts the green stem that developed in the R5 top and R5 whole treatments compared to the proximal non-treated control areas. Different letters indicate significant differences. NTC—non-treated control, R2—full flower, R3—beginning of pod development, R5—beginning of seed development.

### 3.2. Soybean Growth and Yield Formation

The number of pod-bearing nodes is an important yield contributor that is established during vegetative growth [2]. The number of successful flowers depends on the nodal position, as well as the number of nodes that the plant can supply assimilates to; in particular, for flowers in distal positions that are more likely to abscise [39,41,50]. In the present study, soybean NTC plants demonstrated the highest numbers of nodes with 26.5 nodes plant$^{-1}$ in 2019 and 31.0 nodes plant$^{-1}$ in 2020 compared to treatment plants at R2, R3, and R5 stages (Table 3). Plants subjected to flower and pod removal at R2 showed reductions of nearly 8.0 nodes plant$^{-1}$ in 2019 and 11.0 nodes plant$^{-1}$ in 2020 compared to the NTC, and the imposed stress during this stage on both vegetative and reproductive growth may have caused an imbalance (Table 3). The number of mainstem nodes and the plant height continued increasing for indeterminate soybean until R4, and stem elongation generally reached termination at R5, with both stages concurrently occupying the seed-filling period [51]. Across years and treatment stages, on average, plants subjected to imposed stress at R3 had the most nodes plant$^{-1}$ (Table 3) and produced quantities of seeds plant$^{-1}$ that were not significantly different from the NTC (Figure 3b,c). This finding is in agreement with Munier-Jolain et al. [32], who found that soybean plants with greater numbers of nodes showed a lag in physiological maturity that inherently extended the seed-filling period for the whole plant and led to higher yields.

**Table 3.** Main effects of stage and location for observations of plant height and number of mainstem nodes in 2019 and 2020. Different letters indicate significant differences amongst treatments. NTC—non-treated control, R2—full flower, R3—beginning of pod development, R5—beginning of seed development.

| | 2019 | | 2020 | | 2019 | | 2020 | |
|---|---|---|---|---|---|---|---|---|
| *Stage* | Nodes Plant$^{-1}$ | Level of Significance | Nodes Plant$^{-1}$ | Level of Significance | Plant Height (cm) | Level of Significance | Plant Height (cm) | Level of Significance |
| NTC | 26.50 | a | 26.50 | a | 26.50 | a | 31.00 | a |
| R2 | 19.13 | b | 18.50 | b | 26.13 | a | 30.75 | a |
| R3 | 19.38 | b | 22.00 | b | 26.25 | a | 31.25 | a |
| R5 | 22.00 | b | 18.88 | b | 27.13 | a | 30.25 | a |
| *Location* | Nodes Plant$^{-1}$ | Level of Significance | Nodes Plant$^{-1}$ | Level of Significance | Plant Height (cm) | Level of Significance | Plant Height (cm) | Level of Significance |
| NTC | 31.00 | a | 31.00 | a | 26.50 | a | 31.00 | a |
| Middle | 20.00 | b | 19.25 | b | 25.00 | a | 29.00 | a |
| Top | 20.25 | b | 20.75 | b | 26.25 | a | 31.50 | a |
| Whole | 19.50 | b | 18.75 | b | 27.00 | a | 30.25 | a |

Increased dry matter accumulation, or the quantity of total biomass converted to seed biomass, is a primary factor for the harvest index [2,3]. There were no differences in height between plants across treatment stages and locations and no consistent pattern, as plants treated at R5 in 2019 and R3 in 2020 were the tallest, whereas plants subjected to simulated stress at the middle locations were taller in 2019 compared to whole mainstem removal in 2020 (Table 3). Our findings that plant height does not have a relationship with soybean yield recovery are in agreement with previous studies that found no connection between plant size and seed number [52,53].

Contributing factors determining soybean yield include pod number per plant and number of seeds per pod [27]. The number of pods per node results from the balance between the photoperiod and source–sink relationships, with the source—or assimilatory capacity—determining the generation and fate of reproductive structures, which are known as the sink [17,54]. Here, only the 2020 data are discussed (Figure 5). Flower removal at R2 demonstrated yield recovery potential, and R2:M soybean produced −19.3 pods plant$^{-1}$, R2:T produced −5.0 pods plant$^{-1}$, and R2:W produced +5.0 pods plant$^{-1}$, which was not significantly different from the NTC plants that were not subjected to flower removal at R2 (Figure 5). This agrees with previous studies that found that the removal of proximal competing flowers reduced the probability of abscission for distal flowers from 30.0% to 7.0% on

lower mainstem nodes and from 59.0% to 9.0% on upper nodes [39,50]. Soybean's support of a wide range of pods of different ages developing concurrently at R3 can trigger abortion of flowers and development of pods on upper and distal nodes, while the rapid growth of older pods dominates assimilate supply from photosynthesis [54]. Simulation of stress at R3 demonstrated the aforementioned source–sink imbalance, with pod removal at R3:M, R3:T, and R3:W triggering yield losses of 53.8, 53.0, and 66.0 pods plant$^{-1}$, respectively, compared to the NTC (Figure 5). Additionally, these yield imbalances related to the rate and duration of partitioning of dry matter to the seeds, with seed dry matter accumulation being either limited or assisted by assimilate supply during the seed-filling period [51]. When stress was imposed at R5:M and R5:T, there was comparable pod retention until maturity, whereas the R5:W simulated stress losses of 135.5 pods plant$^{-1}$ compared to the NTC could not be recovered (Figure 5). A study simulating soybean response to biomass removal found comparable evidence of a strong correlation between soybean pod density and yield [55].

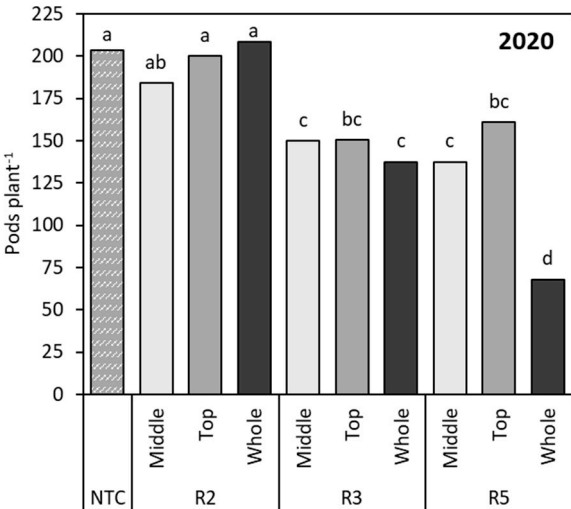

**Figure 5.** Number of pods per plant as affected by cumulative stage and location effects in 2020. Different letters indicate significant differences. NTC—non-treated control, R2—full flower, R3—beginning of pod development, R5—beginning of seed development.

The number of flowers contributing to yield depends on whether the plant produces enough extra flowers to recover lost flowers or pods following a stress event [26]. Flower removal at R2 resulted in the surviving pods demonstrating the lowest number of one-seed pods across years, the largest quantity of two-seed pods in 2020, and the second highest number of three-seed pods and the lowest number of four-seed pods across years (Figure 6a,b). Abscission of flowers due to source–sink imbalances often occurs due to high sink intensity before anthesis and a sink-limited phase following anthesis, which creates a gap in the competitive ability of the reproductive structures to obtain adequate photosynthetic assimilates [50]. For varying locations of flower loss across R2, assimilate resources and energy were concentrated towards the survival of two and three seed pods (Figure 6a,b), which ultimately resulted in seed numbers plant$^{-1}$ comparable to NTC plants (Figure 3a,c). Imbalances in the source–sink relationship can arise from low canopy photosynthesis during demanding periods of reproductive growth but do not affect final seed numbers if the photosynthetic limit is short in duration [26]. The simulated stress at R3 acted as a narrow window in which later-produced flowers had the potential to mitigate yield losses and the soybean focused on fostering resources and growth for primarily one- and two-seed pods across years (Figure 6a,b). Soybean seed size and response to assimilate availability are thought to be co-limited by the pull for resources between the plant as the source and the developing seed as the sink [56]. Surviving pods from R5 plants were consistently lowest for two and three seed pods and highest for four seed

pods (Figure 6a,b), although the four seed pod plants did not fill the substantial yield gap compared to non-stressed soybean (Figure 3a,b). Elgi and Bruening [17] also found that a small number of surviving pods were initiated after R5 in indeterminate cultivars and concluded that, in non-stress environments, the critical period for pod set can extend to R6.

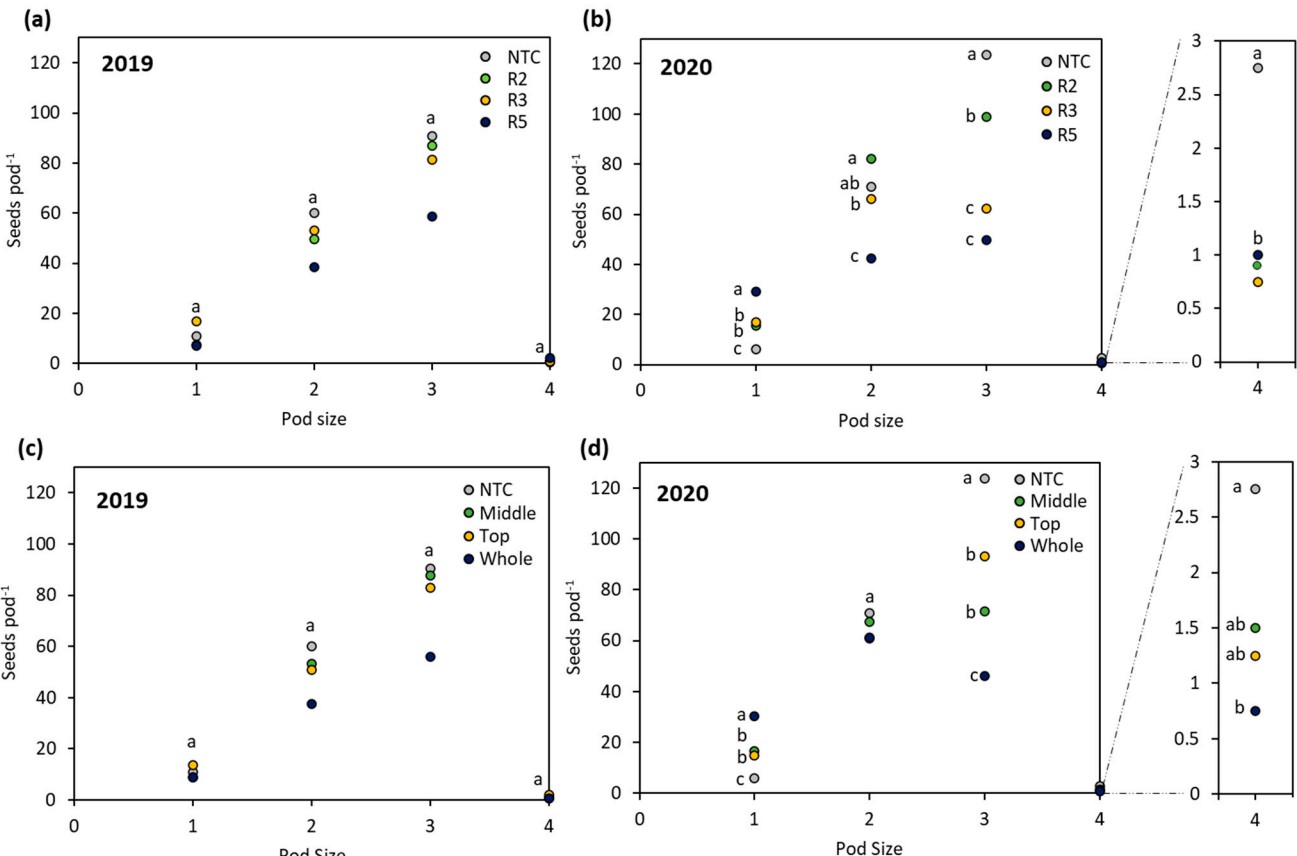

**Figure 6.** (**a**) Comparison of the numbers of one-, two-, three-, and four-seed pods as affected by stage in 2019 and (**b**) 2020. (**c**) Comparison of the numbers of one-, two-, three-, and four-seed pods as affected by location in 2019 and (**d**) 2020. Different letters indicate significant differences and one letter above a set of data points indicates that the level of significance was the same.

Seed numbers at the nodal position were determined at the time assimilatory capacity decreases and seed filling begins, with the risk of seed abortion being highest when dry matter accumulation is limited [53]. The timeframe for the beginning and end of the seed-filling period is dependent upon the location of each reproductive node, as well as the moisture status of the seed [32], which could be interpreted as indicating that yield can be improved by conditions facilitating crop growth for a longer duration [54]. The hypothesis that pods located at the middle of the mainstem act as the highest yield providers held true for consistent contributions of one, two, three, and four seeds per pod in 2019 and 2020 (Figure 6c,d). For the same node position, the seed-filing period was quantified as having a 10 to 25 day gap between the first and last surviving pod, which could help explain why the mainstem location, the most continually productive reproductive region of the plant, comprised the widest range of seeds per pod numbers [17,22,27]. The number of seeds per pod for the top location followed a similar trend, with no significant differences compared to the mainstem location in 2019 and 2020 (Figure 6c,d). Previous research on removing pods from proximal node positions found that the number of seeds per pod at distal node positions increased from 2.5 seeds per pod with proximal pods present to 2.7 seeds per pod for distal nodes when proximal pods that would otherwise have been in competition were not present [11]. These findings together indicate how the number of seeds per pod in the

top portion of the mainstem can be comparatively productive for the middle portion of the mainstem when the large sink of mainstem pods is removed and assimilates are available to supply pods at distal positions [11]. Sink competition for assimilates was substantially decreased with whole removal of flowers or pods, allotting resources for the production and survival of the one-seed pods that compromised the majority of yield contributions in 2019 and doubling the number of one-seed pods produced by top and middle treatments, as well as tripling in comparison to the NTC in 2020 (Figure 6c,d). This was in agreement with the pod removal treatments studied by Prolux and Naeve [56], which caused sink-limited yield as a result of decreasing seed numbers while, at the same time, boosting the seed size of surviving pods with a surplus in assimilate supply.

### 3.3. Potential for Pod Recovery

As the seed variety used in this study was an indeterminate variety, it should be noted that there are differences in the limitations affecting mainstem and branch growth between indeterminate and determinate types during reproduction. Determinate plants cease vegetative growth before entering flowering, while the vegetative growth and reproductive growth phases of indeterminate plants overlap, allowing continued mainstem and branch growth [57,58]. The three-way interaction between stage, location, and nodal position on both the mainstem and branch and its influence on the number of pods contributing to yield were of interest (Table 4). Beginning with the number of pods on the mainstem, R2:M had +14.2 pods and −5.0 pods compared to the NTC in 2019 and 2020, respectively (Table 4). Continued production of mainstem nodes in indeterminate cultivars has been more closely linked with increased plant productivity through the increased window for flower production creating more surviving pods rather than the availability of assimilates [17]. Numbers of pods located on the mainstem at the R3 stage were highest for R3:M in 2019 and R3:T in 2020, although they were not significantly different, while the number of pods at R5:T was greatest at the R5 stage in 2019 and 2020 (Table 4).

**Table 4.** Effects of the interaction between stage, location, and position on the number of pods contributing to yield in 2019 and 2020. Different letters indicate significant differences amongst treatments. NTC—non-treated control, R2—full flower, R3—beginning of pod development, R5—beginning of seed development.

| | | 2019 | | | | 2020 | | | |
|---|---|---|---|---|---|---|---|---|---|
| Stage | Location | Position: Mainstem | Level of Significance | Position: Branch | Level of Significance | Position: Mainstem | Level of Significance | Position: Branch | Level of Significance |
| NTC | NTC | 17.35 | fg | 44.71 | a | 64.25 | fg | 110.25 | b |
| R2 | Middle | 31.50 | c–e | 41.52 | ab | 59.27 | g–i | 109.83 | bc |
| R2 | Top | 24.58 | e–g | 35.91 | b–d | 60.17 | gh | 103.83 | b–d |
| R2 | Whole | 16.04 | g | 39.29 | a–c | 49.92 | Ij | 133.25 | a |
| R3 | Middle | 28.35 | d–f | 42.91 | ab | 48.33 | J | 104.33 | b–d |
| R3 | Top | 15.83 | g | 40.4 | a–c | 52.33 | h–j | 99.50 | d |
| R3 | Whole | 24.75 | e–g | 30.833 | c–e | 45.42 | J | 100.83 | cd |
| R5 | Middle | 15.65 | d–f | 22.87 | e–g | 34.33 | K | 75.75 | e |
| R5 | Top | 18.35 | fg | 23.92 | e–g | 52.67 | h–j | 71.83 | ef |
| R5 | Whole | 17.75 | fg | 2.83 | h | 13.75 | l | 7.58 | l |

The temporal patterns for flower and pod production for the mainstem and branch nodes followed a similar pattern, and it is thought that synchronous flowering and pod set can balance the amounts of assimilates distributed, resulting in more equally weighted sinks for reproductive structures at similar stages [17]. However, branch yield often shows stable performance across environments due to genetics and the vegetative and reproductive growth of branches, and branch seed yield is understood to be more sensitive to moisture stress conditions than mainstem pods. Thus, the different environmental conditions in 2019 and 2020 could be the primary factors affecting the variation from year to year [24,27,59]. Overall, optimal branch vegetative growth is indicative of high seed yields [24], and it was found that, across years, stages, and locations, most of the pods contributing to yield were pods on branches, except for in the R5:W treatment, which had more pods located on the mainstem (Table 4 and Figure 7a,b). This was in agreement

with previous studies reporting that branch seed yield is often higher than mainstem seed yield [27,60]. It is assumed that compensation of yield through increased branch development and pod production per plant is primarily due to higher quantities of branch reproductive nodes rather than increases in pod loading within nodal positions [42,60,61]. Branch pods from R2:W soybean were not significantly different from the NTC, showing differences of $-5.4$ pods plant$^{-1}$ in 2019 and $+23.0$ pods plant$^{-1}$ in 2020 (Table 4), which was taken as evidence of branch pods governing yield more strongly than mainstem pods across years (Figure 7a,b). Similarly, R3:M was not significantly different from the NTC, showing differences of $-1.8$ pods plant$^{-1}$ in 2019 and $-5.1$ pods plant$^{-1}$ in 2020 (Table 4). However, the significantly lower numbers of branch pods in comparison to mainstem pods in 2019 and 2020 indicated there was a comparatively higher number of mainstem pods that also contributed to yield (Figure 7a,b). At R5, removal of pods at R5:M and R5:T resulted in no significant differences between the two treatments but reduced the pod load by approximately 21.3 pods plant$^{-1}$ in 2019 and 36.5 pods plant$^{-1}$ in 2020, while the whole treatment retained the lowest pod load (Table 4).

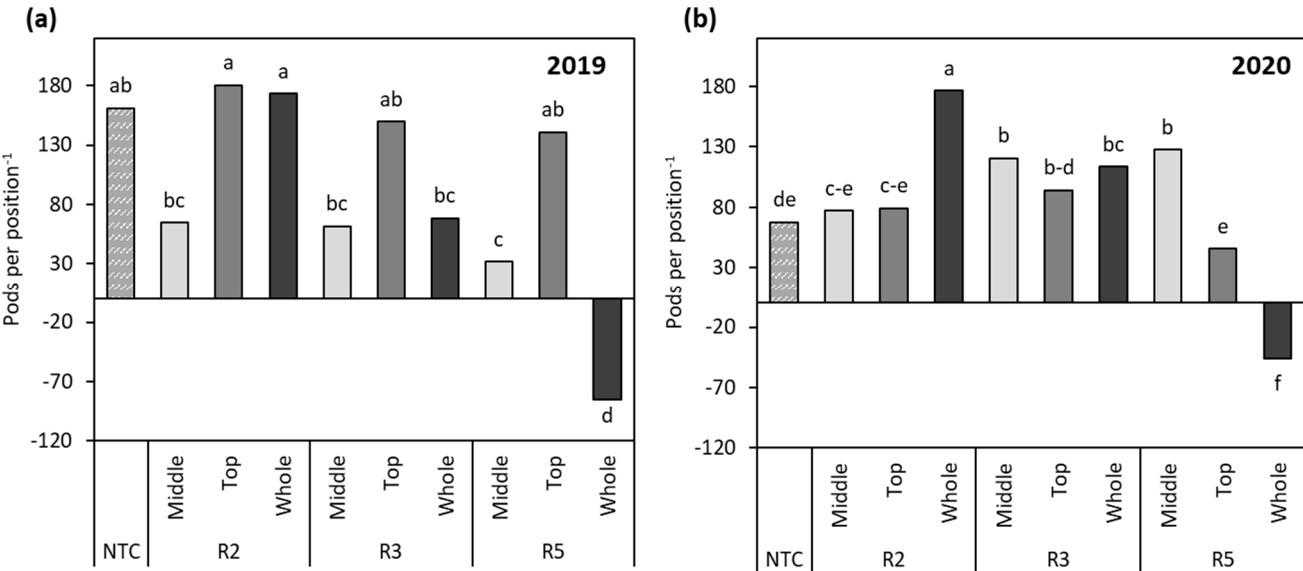

**Figure 7.** (**a**) Comparison of the changes in mainstem versus branch (MVB) pod positions as influenced by stage and location of simulated stress in 2019 and (**b**) 2020. A positive number indicates that more pods were located on the branches, while a negative one signifies that pod numbers were concentrated on the mainstem. Different letters indicate significant differences. NTC—non-treated control, R2—full flower, R3—beginning of pod development, R5—beginning of seed development.

Branch growth has been reported to arise from photoperiodic effects [62,63], but branch node differentiation has also been hypothesized to be promoted by other factors, such as a longer growth period facilitated by increased solar radiation [54]. Soybean under irrigation has been shown to produce greater yield from the branches in comparison to dryland soybean, resulting in the idea of increasing soybean branching as a method to increase yields even in stressful conditions [24,27]. However, the susceptibility of branch growth to moisture stress at reproductive stages principally limits yield through reduced branch growth, thus resulting in fewer yield-bearing branch nodes [27].

## 4. Summary

From the present study, it can be concluded that the removal of reproductive structures during the early-season stages R2 (full flower) and R3 (beginning of pod development) had minimal impact on yield. Additionally, increased likelihood of yield recovery was dependent on the removal event occurring early on during reproductive growth, as the loss of structures at any magnitude during R5 was detrimental to yield. The impacts of location

and potential recovery were highly dependent on the timing of pod removal. Soybean subjected to the R2 treatments responded across removal locations and within years to early-season loss by increasing flower production on branches and recovering flower production on the mainstem, ultimately mitigating yield loss. Physiological responses to pod removal from the mainstem at R3 did not differ, with flower and pod removal from the middle portion of the mainstem producing similar responses. Similarly to R2, the asynchronous soybean flowering period demonstrated the ability to recover and retain lost mainstem reproductive structures through the R3 growth stage. The location where a loss event was experienced became increasingly important at the R5 growth stage (beginning of seed development). This is because the middle portion of the mainstem is a large sink and major contributor to the yield components seed number and seed weight, as seen from the impact of both middle and whole pod removal. In summary, the impact of stress on yield was minimal at R2, increased at R3, and was greatest at R5, with increasing levels of pod removal concomitantly reducing the seed number sink. Overall, this study shows that soybean plant response to flower/pod removal, similar to that experienced in a stress event, involves an interactive process that depends on when the event occurred and whether the event was isolated to a portion of the plant or extended across the entire plant.

**Author Contributions:** Conceptualization, J.L.; methodology, S.K. and J.L.; validation, S.K., A.B. and J.L.; formal analysis, S.K. and J.L.; investigation, S.K., A.B. and J.L.; data curation, S.K., R.S. and J.L.; writing—original draft preparation, S.K.; visualization, S.K., V.K. and S.S.; supervision, J.L.; project administration, J.L.; funding acquisition, J.L. All authors worked on editing and final manuscript preparation. All authors have read and agreed to the published version of the manuscript.

**Funding:** Funding for this project was provided by the Oklahoma Soybean Board. The authors would like to thank them and the growers for their support for the project.

**Data Availability Statement:** Data are available in the manuscript or upon request from the corresponding author.

**Conflicts of Interest:** The authors declare no conflict of interest.

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
