# Peer review of "Response of Soybean Yield and Certain Growth Parameters to Simulated Reproductive Structure Removal"

_agronomy, doi:10.3390/agronomy13030927_

Round 1

Reviewer 1 Report

It is an excellent effort to report issues related to sustainable Soybean growing and development in climate change scenarios. 

But there are some issues that hinder me from favouring this article for publishing.

This article is derived from and reports the results of the author's Master's degree dissertation that is already been published online with the same title, which could be viewed at "https://shareok.org/handle/11244/325475". 

If considered for publication by the editorial office, the following points are suggested for improvement.

1. Abstract is not in a stand-alone standard format that can explain the research to individual readers. It needs to improve by properly explaining the research background and methods and support with significant quantitative results. Also, explain the impacts of this study on future research and developments.

2. Define all abbreviations like "OK, T, M, W, K and P...." etc. and variables like growth stages R1, R2.....Rn and others if any at their first use in all three sections; abstract, body and conclusion.

3. Some grammatical corrections are required throughout the article.

4. Figure captions and nomenclature should be according to Journal's format, the caption should be below/next to the figure.

5. Formula equation at line 205, complete it in equation format.

6. Results data in Table 2 is not complete and well elaborated.

7. In methodology, properly and clearly define and describe treatments that were applied in the study (main treatment & sub-treatment).

8. Conclusion should also be well structured and elaborated by quantitative results.

Reviewer 2 Report

This field study evaluates yield recovery from removal of reproductive structures (as a result of biotic and abiotic stressors) in Soybean in 3 site-years in Ok, USA. This is a very interesting experimental study with an appropriate research design and clearly presented results that support conclusions important for producers and global food and feed security. The use of English, apart from some issues noted in the attached manuscript, is fine. For these reasons I recommend the publication of this study.

Reviewer 3 Report

Comments:

1. L 114 and Figure 1b: It is not a mistake to give rainfall totals in cm, but in publications their amount is most often specified in mm. I leave to the Authors' consideration the possible conversion of cm units to mm.

2. L 115: In 2020, Bixby received 92.5 m of rainfall (meters ??).

3. Please provide the name of the soybean cultivar included in the study.

4. The Authors provided only the plant density of soybean plants per unit area. Please also provide other sowing parameters like row spacing, depth of seed sowing.

5. The description of the way of conducting the experiment (experimental design) needs to be improved, because in my opinion it is not clear:

a) how many plots were there in the experiment?

b) the plot area was 9.15m2 (6.1m x 1.5m) which was divided into smaller target areas (3.0m2; 3.1m x 0.8m)?

c) on what area of the plot and on how many plants abortion treatments of reproductive structures were performed (in different parts of plants and in three developmental stages)?

d) on how many plants in total for each treatment level were biometric measurements performed? Assuming that 3 plants were randomly selected from each plot for each treatment level x 4 replicates, this is only 12 plants for each treatment level. If so, in my opinion the sample size is too small to draw reliable conclusions.

6) L 574: Editorial error - part of the sentence was deleted?

7) The Conclusion chapter needs significant shortening.

Reviewer 4 Report

In my opinion, a very good manuscript requiring only few corrections before publishing.

Line 115 Figure 1a present the course of the temperature, not rainfall data

Lines 115-116 Sentence need correction "In 2020, Bixby received 92.5 m of rainfall and was the sole established location, due to planting failure at Perkins from early-season flooded conditions (Figure 1a)." Inappropriate rainfall units in Bixby (mm not m). For what period was the sum of rainfall given?

Line 117 Figure 1b present rainfall data, not temperature.

Line137/138 Wrong bacteria latin name? "Bradyrhizobium rhizobium japonicum"? My opinion should be "Bradyrhizobium japonicum

Line 176 Sentence "across site years, all treatments and interactive effectes were replicated four times" I don't understand that. Please precise.

Figure 7. - covers partly the text on page 16

Table 4 - 2019 and 2020 header are outside the table. I suggest putting in the table.

Round 2

Reviewer 1 Report

The article from the thesis is ok to publish in general. The editorial office will decide on this article.

I suggest modifying the title of this article somehow to make it different from the thesis but precisely relevant to the content and cite the thesis in this article appropriately.

I am not satisfied with the Abstract and again I suggest modifying and improving the Abstract according to Agronomy Journal Guidelines for authors.

Editorial mistakes would be corrected by the editorial office.

The conclusion is not mandatory for this journal, if it is being included, it should be concisely-structured and supported by the reported results.

It is preferred and recommended to submit a Word File of the article for review.

Author Response

Response of soybean yield and certain growth parameters to simulated reproductive structure removal.

Response to reviewer_2

  • This is a good point and I am glad the reviewers raised it. The title has been changed to a more focused title based on data presented.  Also, the fact that this is chapter from a thesis has been noted for full disclosure. Additionally, citations was corrected accordingly.
  • The authors have done a somewhat complete overhaul in order to attempt to change the structure in accordance to the reviewers request. We have taken some of the comments made from the initial review and additional comments to try to address these issues within the narrow word count.  Hopefully, this adds enough information to suit the reviewers request.
  • The authors did not see that a conclusion section was not required. Therefore, we have taken the reviewers comments and changed this moderately.  The authors do agree this functioned more as a summary of concepts and ideas.  Therefore, if the journal will allow we have changed this to a summary section and made significant changes to attempt to address some of the issues the reviewer has made.  Again, hopefully this will satisfy the requests for the reviewer.
  • The authors did submit in a word file. Maybe the editor can clarify the issue here.  Anything the authors can do to help, please feel free to request.

Reviewer 3 Report

Dear Authors,

Most of my comments were taken into account when the article was finalized. However, I have one more comment:

L 171. This table should be in L 201 - after its title „TABLE 1 Planting population, planting date, and harvest date for field trials conducted in Perkins, OK in 2019 and 159 Bixby, OK in 2019-2020.”

Best regards,

Reviewer

Author Response

The suggestion was corrected.  Hopefully in a more finalized version several of the inconsistencies associated with formatting can be further addressed. 

Thanks you for your comments.